# Relationship Between Breed, *GH* and *CAST* Genotypes, and FA Composition in the Ovine Intramuscular Fat of *Musculus Semimembranosus*

**DOI:** 10.3390/ani15202992

**Published:** 2025-10-15

**Authors:** Evaldas Šlyžius, Gintarė Zaborskienė, Vaida Andrulevičiūtė, Ingrida Sinkevičienė, Vilija Buckiūnienė, Renata Bižienė

**Affiliations:** 1Department of Animal Breeding, Faculty of Animal Sciences, Lithuanian University of Health Sciences, LT-44307 Kaunas, Lithuania; evaldas.slyzius@lsmu.lt; 2Department of Food Safety and Quality, Faculty of Veterinary Medicine, Lithuanian University of Health Sciences, LT-44307 Kaunas, Lithuania; gintare.zaborskiene@lsmu.lt; 3Department of Biochemistry, Faculty of Medicine, Lithuanian University of Health Sciences, LT-44307 Kaunas, Lithuania; vaida.andruleviciute@lsmu.lt (V.A.); ingrida.sinkeviciene@lsmu.lt (I.S.); 4Institute of Animal Rearing Technologies, Faculty of Animal Sciences, Lithuanian University of Health Sciences, LT-44307 Kaunas, Lithuania; vilija.buckiuniene@lsmu.lt; 5Institute of Biological Systems and Genetic Research, Faculty of Animal Sciences, Lithuanian University of Health Sciences, LT-44307 Kaunas, Lithuania

**Keywords:** sheep, gene variants, fatty acid, *GH* gene, *CAST* gene, growth

## Abstract

This study examined how different sheep breeds and genetic variants affect the fatty acid composition of lamb and animal growth performance. Lithuanian Black-Headed lambs had healthier meat, containing more beneficial fatty acids (omega-3 and polyunsaturated fatty acids) and less saturated fat compared to crossbreds. Lambs with the MN genotype of the *CAST* gene grew better and had healthier fat profiles than lambs with the MM genotype. The BB genotype of the *GH* gene resulted in the highest growth rates. However, lambs with the AA genotype had the healthiest fat profile, with lower levels of saturated fat and higher levels of polyunsaturated fat. Crossbreeding strategies and genetic selection could be used to make lamb grow faster and produce healthier meat.

## 1. Introduction

Livestock production is an essential component of agricultural production, as livestock consume natural agricultural resources in animal protein production [1,2,3,4]. Livestock production represents about 30% of the total agricultural income and is important for achieving food security, especially from animal protein [5,6,7]. Native breeds are well adapted to agricultural climatic conditions, resistant to many tropical diseases, and can survive and produce milk on poor forage and fodder resources [7,8]. Some of these breeds are known for their high milk and fat production. Agriculture faces a great challenge due to the increasing global population, which requires increasing food production while implementing effective and sustainable production techniques. The ability of ruminants, especially small ruminants, to use low-value inputs and produce high-value outputs is well-known [9].

Growth traits are economically significant in sheep breeding, influenced by multiple genes governing growth, bone formation, birth weight, weaning weight, and muscle development. Growth hormone, a 191-amino acid polypeptide secreted by the somatotropic cells of the anterior pituitary gland, is encoded by the *GH* (growth hormone) gene [10,11]. The ovine growth hormone gene is found on chromosome 11 (11q25), consisting of five exons, 1795 bp in size. *GH* is pivotal for postnatal growth, tissue development, lactation, reproduction, and the metabolism of proteins, lipids, and carbohydrates [12]. There have been reported associations between polymorphic variants of the *GH* gene and production traits such as live weight [13,14,15], body measurements [16,17], birth weight [18,19], and weaning weight [20,21,22]. Given that one of the main functions of growth hormone is to promote body and muscle development, the *GH* gene was selected as a candidate marker in this study [16].

The calpastatin gene (*CAST*) is important for regulation of meat tenderization, birth weight, and weaning weight. The *CAST* gene is found on chromosome 5 (5q15) and consists of 29 exons, 89,553 bp in size.

Many researchers have determined differences in growth rate, fat deposition, and productivity and their dependence on the diet, species, breed, and gender of lambs [23,24,25,26,27]. However, due to a poor selection method, these animals’ productive capacity has deteriorated over time [28]. Moreover, genetic studies on sheep breeds have revealed gene candidates associated with productivity [29]. The presence of *GH* and Calpastatin (*CAST*) influence growth rate [30,31]. The height of degree of heterozygosity of *GH/HaeIII* polymorphisms can be a genetic marker in a sheep selection programme for improvement of growth rate and weaning weight.

Growth hormone (*GH*) and Calpastatin (*CAST*) gene polymorphisms can affect not only productivity but also meat fat composition, which affects not only meat’s sensory properties but also physiological benefits for human health.

Lamb meat is rich in fatty acids (FAs). Also, the proportions of saturated fatty acid (SFA), monounsaturated fatty acid (MUFA), and polyunsaturated fatty acid (PUFA) in lamb meat are of great significance for maintaining normal physiological function in the human body [32,33]. The flavour of lamb perceived by consumers is generally closely related to the content of certain beneficial FAs [34,35].

Ruminant meat is a source of proteins and minerals for human nutrition. However, high levels of saturated fatty acids (SFAs) in livestock meat are associated with chronic and heart diseases [36]. Many efforts have been made to improve animal fat content in terms of management and genetic improvement [37,38]. Intramuscular fat contributes to tenderness and can be affected by selection. Many polymorphisms of the *CAST* gene have been related to meat quality indicators and characteristics, such as tenderness, colour, and intramuscular fat (IMF) content in cattle and pig carcasses [39,40,41,42,43,44]. It has been reported that several genes related to lipid metabolism regulate the composition of FAs in ruminant meats [45]. The FA composition of lamb meat is an important trait for the determination of flavour and nutrition, and this meat can provide essential FAs for human health [46]. The types and levels of FAs are influenced by polygenic and environmental factors [47,48]. A better knowledge of the molecular architecture of meat FA composition is important as it may generate new opportunities for more effective marker-assisted breeding, leading to economic benefits for the sheep industry [49]. Nevertheless, compared to beef and pork, the studies of genetic effects, including the mutation and expression of candidate genes, on the FA composition of lamb are few. Calpastatin (*CAST*) is involved in the calpain–calpastatin system, which influences many important processes, including muscle development and growth [30], and is also known to regulate the degradation of myofibrillar proteins both in living and in post-mortem muscle tissue [50,51].

Recently, a special interest in developing and improving native breeds has increased. The important role of these genes in animal growth is well-known, so their polymorphisms and interactions with growth and meat characteristics are likely to be markers in a sheep breeding programme to improve productivity in particular. However, research focusing on the relationship between the *CAST* or *GH* genes and meat quality, especially FA composition, in sheep meat is limited. Since the *GH* gene affects lipid synthesis and mobilization, its genetic variations can determine which fatty acids are stored in the muscles. The *CAST* gene regulates muscle protein degradation and structural changes and therefore affects meat texture. By studying the polymorphisms of these genes, it is possible to identify the genetic factor that determines a healthier and higher-quality fatty acid profile of meat, and the results obtained can be applied in breeding to improve the nutritional value of meat and provide economic production value. Considering the lack of studies on the influence of these gene polymorphisms on the fat quality of ovine, the association of *GH* and *CAST* gene polymorphisms with the fatty acid profile of meat will be determined. The aim of this study was to investigate the relationship between breed, *GH* and *CAST* genotypes, and FA composition in the ovine intramuscular fat of *musculus Semimembranosus*. 

## 2. Materials and Methods

This study was conducted at the sheep farm, located in Lithuania (55°46′06.2″ N 23°44′08.4″ E). A total of 175 blood samples were drawn by jugular vein puncture from Lithuanian Black-Headed (N43), Lithuanian Black-Headed *Ile de France (N43), Lithuanian Black-Headed *Suffolk (N44), and Lithuanian Black-Headed *Texel (N45) lambs from the 2024 season. The meat of the studied sheep breeds is characterized by good muscle mass and flavour: Lithuanian Black-Headed meat is medium-fat and tender, Ile de France—lean, with a high percentage of muscle yield, Suffolk—juicy and tender, and Texel—extremely lean, very muscular, and firm.

Two feeding systems were applied depending on the season: indoor feeding (late November to early May) and pasture-based feeding (mid-May to early November). During the indoor period, lambs received hay, haylage, 0.3–0.4 kg of concentrated feed per day, and mineral licks; before weaning, additional protein-rich feed (0.1–0.2 kg per lamb) was provided. Pasture-based feeding consisted of ad libitum fresh grass, water, and mineral licks. After weaning, female lambs grazed on cultivated pastures, while ram lambs remained indoors and were fed hay, haylage, and concentrated feed containing a pea–oat–vetch mixture, barley, and protein supplements.

The lambs were born in January, and weaning took place in May. Approximately 30% of the lambs were from twin births.

The lambs were weighed weekly, and the growth rate was calculated using birth weight and month weight values.

The investigation was carried out in the Lithuanian University of Health Sciences, K. Janušauskas Laboratory of Genetic (Kaunas, Lithuania). In the present study, 175 animals were tested. Genomic DNA was extracted from blood samples taken into EDTA containing tubes, using a “GeneJET Genomic DNA Purification Kit” (Thermo Scientific, Waltham, MA, USA).

The methods of polymerase chain reaction and restriction length polymorphism were used to genotype growth hormone (*GH*) (C → G substitution) and (*CAST*) (A → G substitution at position 286) gene polymorphisms [52,53]. The characteristics of the primers used for amplification and restriction analysis are presented in Table 1.

The fragment sizes were determined by agarose gel electrophoresis in 2% agarose gel (Thermo Scientific, Waltham, MA, USA), stained with ethidium bromide (Thermo Scientific, Waltham, MA, USA). Fragment identification was performed in ultraviolet light, using a MiniBIS Pro Video Documentation System (DNR Bio Imaging System, Neve Yamin, Israel).

Meat samples of four lamb breeds were taken for analysis: Lithuanian Black-Headed (N43), Lithuanian Black-Headed *Ile de France (N43), Lithuanian Black-Headed *Suffolk (N44), and Lithuanian Black-Headed *Texel (N45), from the slaughterhouse (X) located in Radviliškis district, Lithuania. The samples of 200–250 g from *musculus. Semimembranosus* were taken 48 h after carcass meat cooling. The samples were kept in a refrigerator at +4 °C temperature 24 h until the homogenization and extraction of intramuscular fat. Fatty acids extracted with n-hexane were methylated with a 2 mol/L KOH solution in anhydrous methanol.

The fatty acid methyl esters (FAMEs) were analyzed using a gas chromatography mass spectrometry (GC/MS) instrumental technique, in accordance with LST EN ISO 12966—2:2017 [54]. Chromatographic analysis of fatty acid methyl esters was performed using a PerkinElmer Clarus 680 gas chromatograph and a PerkinElmer Clarus SQ8T mass spectrometer with a capillary column SP-2560, 100 m × 0.25 mm × 0.20 µm. Conditions for chromatographic analysis were as follows: the injector and detector temperatures were maintained at 250 °C; the oven temperature was initially 60 °C for 1 min and was increased to 180 °C at a rate of 12 °C/min and held for 30 min. The spectrometer temperature mode was 5 °C/min. up to 300 °C, holding for 2 min. The samples of FAME in hexane (1.0 μL) were injected through the split injection port (10:1). Carrier gas (He) flow rate was 1 mL/min. A Supelco 37 Component FAME Mix fatty acid kit (Merck, Darmstadt, Germany) was used for fatty acid identification. Individual FAMEs were identified by comparing their retention times with those of the standard, and the results were calculated as a percentage of the total FAMEs (% of TFA). Individual FAs were used to calculate the sums of saturated fatty acids (SFAs), monounsaturated fatty acids (MUFAs), and polyunsaturated fatty acids (PUFAs).

Statistical data analysis was performed using SPSS 25.0 analytical software (IBM Corp., Armonk, New York, NY, USA). The data were presented using descriptive statistics. Normal distributions of variables were assessed using the Kolmogorov–Smirnov test. One-way analysis of variance (ANOVA) was used to determine statistically significant differences between the means. Multiple comparisons of groups mean were calculated using the post hoc Tukey test. The differences were considered significant at *p* < 0.05.

## 3. Results and Discussion

### 3.1. Fatty Acid Profiles According to Sheep Breed in the Musculus Semimembranosus

The fatty acid profile according to sheep breed is presented in Table 2. Some essential differences were noticed among the breeds. For example, the meat of Lithuanian Black-Headed showed the lowest amount of SFA (mainly C16:0, and C18:0) and the biggest amount of PUFA. This could be associated with increased amounts of FAs ω-3 (C22:6n6, C20:5n3 and 18:3n3) and ω-6 (C18:2, C20:4). Significant differences in SFA, PUFA, FA ω-3 and ω-6 (*p* < 0.001) were characteristic of all three other breeds compared to Lithuanian Black-Headed. The meat of free breeds (Lithuanian Black-Headed, Lithuanian Black-Headed* Suffolk, and Lithuanian Black-Headed) showed the lowest content of trans fatty acids (trans C18:1 and trans C18:2). The fatty acid profiles between purebred and crossbred lamb fats were characteristic of the case of the Latvia Darkhead breed crossed with other popular breeds [55]. In contrast, the fatty acid profiles differ significantly in the meat of Artli, Hemsin, Cepni, Karayaka, and OF lambs [56]; for example, the SFA and MUFA contents are 41.9 ± 1.2% of TFA and 41.3 ± 0.61% of TFA in Artli and 60.2 ± 1.87% of TFA and 25.1 ± 1.76% of TFA in OF, respectively. Of course, these results might be influenced by the well-known and investigated nutritional impact of the differences in FA profiles [57].

The ratio ω-6/ω-3 FAs varied from 2.71 ± 0.31 (Lithuanian Black-Headed * Suffolk) to 4.52 ± 0.24 (Lithuanian Black-Headed) and did not reach the desirable values of betwen 1 and 2, according to [58]. These values are similar to those obtained by other researchers [59,60,61]. A slight decrease from 3.72 in the control group (Romanov × Romanov) to 3.5 in the experimental group (Romanov × Hisar (experimental group) was noticed in [61] and was shown to be promising for reducing the risk of chronic diseases. The much bigger ratio in the range 14.43–13.23 was determined to be the case in pure Lori-Bakhtiari fat-tailed sheep and their crosses with the Romanov-tailed breed [62].

### 3.2. Influence of Different Breeds on Lamb Weights

In Lithuania, the genetic improvement of local sheep breeds for meat production is slow, and data on the productivity of crossbred offspring is lacking. However, it can be stated that the majority of the lamb sold in Lithuania is obtained by crossing purebred rams with local breeds of sheep that are resistant to local conditions but have low productivity. Purebred local lambs had the highest birth weight, but they gained less weight during growth. Overall, it can be observed that Lithuanian Black-Headed lambs crossbred with other meat breeds reached higher body weights, even though they were fed under the same feeding conditions as purebred lambs. Crossbred Lithuanian Black-Headed/Suffolk lambs had the highest weight gain; by the end of the follow-up, the crossbreeds had a higher weight compared to the purebred Lithuanian Black-Headed lambs (Table 3). These results are consistent with results reported by Popa et al. [63], who reported that Suffolk × Turcana crossbred lambs weaned at the age of 90 days register heavier weights of 25.52 kg in males and 23.87 kg in females, compared with purebred Turcana lambs, which register average weights of 18.58 kg in males and 16.69 kg in females. The findings of Mohammad et al. [64] indicated similar differences; crossbred lambs demonstrated superior growth parameters, such as elevated birth weight, overall body weight gain, and enhanced average daily weight gain. According to these studies, the process of crossbreeding provides substantial economic advantages in the region by enhancing the quantity of meat production. 

### 3.3. Influence of Different Breeds on Lamb Genetic Testing Results

After genetic testing, the results showed a 622 bp fragment of the *CAST* gene. After digestion with MspI restriction endonuclease, two fragments are produced—336 bp and 286 bp (M allele) (Figure 1). Two genetic variants (M and N) and two genotypes (MN and MM) were found. The genotypes frequencies of the MM and MN genotypes were 0.57 and 0.43, respectively. Many researchers have also found only two genotypes of the CAST gene prevalent among sheep. Kolosov found the presence of two genotypes MM and MN to be established with a frequency of 0.88 and 0.12, respectively [65]. The NN genotype was also not determined in Uppe et al. They found MM and MN genotypes with frequencies of 0.92 and 0.8, respectively. 

Digestion with HaeIII showed 10 restriction sites on A allele and 11 sites on B allele of the *GH* gene (Figure 2). Two genetic variants (A and B) and three genotypes (AA, AB, and BB) were found. The genotypes frequencies of the AA, AB, and BB genotypes were 0.32, 0.35, and 0.33, respectively. A slightly different frequency distribution of alluvial deposits was found by Salah A. El-Mansy [66]. They found that genotype frequencies were AA—0.2, AB—0.33, and BB—0.47. Sunil Kumar found two genotypes—AA (0.62) and AB (0.38)—in the studied population [67].

### 3.4. Influence of Birth Weight, Weight Gain, and Fatty Acids Composition of Different Genotypes of the CAST Gene

The effects of the *CAST* gene on birth weight and weight gain are illustrated in Table 4. The obtained data shows that lambs with the MN genotype had higher birth weight and weight gain. The same result was given by researchers studying Pakistani breeds, where lambs carrying the heterozygous MN genotype demonstrated higher weight gains [68]. Among Awassi sheep from Jordan, individuals with the MN genotype showed higher final body weights, increased daily weight gain, and improved feed efficiency. Similarly, West Siberian mutton sheep carrying the MN genotype were found to have a 6.9% greater average daily weight gain from birth to weaning and a 20% higher live weight [69].

The influences on the FA profile of genotypes were investigated. The results of *CAST* gene MN and MM types are presented in Table 5. The investigated values of SFA, MUFA, PUFA, DFA were very similar, there is only a slight difference for the *CAST* gene observed in the content of FA ω-3 (1.86 ± 0.20% of TFA for MM and 3.03 ± 0.41% of TFA for MN).

### 3.5. Influence of Birth Weight, Weight Gain, and Fatty Acids Composition of Different Genotypes of the GH Gene 

After analyzing the influence of *GH* gene genotypes on birth weight and average daily gain, the best results show lambs with the BB genotype: 4.78 ± 0.11 and 0.267 ± 0.01, respectively (Table 6).

The results of gene *GH* genotypes on fatty acids composition are presented in Table 7. The *GH* gene genotype showed a significant influence on the fatty acid profile. For instance, SFA varies from 40.98 ± 0.51% of TFA (AA) to 50.84 ± 0.49% of TFA (AB) and PUFA varies from 11.54 ± 0.75% of TFA (AA) to 8.94 ± 0.86% of TFA (BB). The contents of FAs ω-3 and ω-6 are similar in all samples, and do not differ between genotypes. The AA genotype shows the lowest SFA content and the highest PUFA content. The BB genotype has the highest DFA, which is beneficial, but also relatively high SFA. The AB genotype is intermediate in most traits but has the highest trans fats and ω -6/ω -3 ratio, which may be less desirable (Table 7). As Ribeiro et al. note, animals that produce leaner meat tend to have higher levels of polyunsaturated fatty acids (PUFAs), likely due to a greater proportion of membrane lipids [70]. Based on this, the minimal impact of genotype on most of the detected fatty acids—including PUFAs—as well as on their total content and mutual ratios, may be explained by the similar fatness levels observed among the evaluated genetic groups, as also reported by Vargas et al. [71]. The analysis of fatty acid profiles in relation to *GH* gene polymorphism revealed notable differences among the genotypes. The GH-AA genotype appears to be the most advantageous from a nutritional perspective, as it was associated with a lower proportion of saturated fatty acids (SFA), higher levels of polyunsaturated fatty acids (PUFA), and a more favourable omega-6 to omega-3 (ω-6/ω-3) ratio. These traits are indicative of leaner and healthier meat, which is highly desirable from both consumer health and product quality standpoints. In contrast, the GH-BB genotype, while showing the highest levels of desirable fatty acids (DFAs), was also characterized by a relatively high SFA content, which may reduce its overall nutritional value. Nonetheless, its elevated DFA concentration suggests potential benefits for specific quality attributes such as flavour or oxidative stability. The GH-AB genotype demonstrated the least favourable lipid profile, with increased trans fatty acid content and a higher ω-6/ω-3 ratio. These factors are generally associated with a less desirable fatty acid composition, which could have negative implications for meat quality and human health. Overall, these findings suggest that *GH* gene polymorphism plays a significant role in shaping the fatty acid composition of lamb meat, and that selection for the AA genotype may contribute to improving the nutritional quality of lamb products.

In addition, there are other factors that contribute to FA content, such as sex [72], slaughter age [73], and, especially, feed [74]. It is important to note that besides the effects of breeds (including genetic factors), grassland types (including environment and climate) and grazing habits (including local latitude and culture) affect the meat FA content of local sheep breeds in a natural grazing system, Non-genetic factors have important effects on the economic traits of lamb. Nevertheless, genetic effects, including major genes, are more efficient for improving the meat quality in local sheep breeds. Based on the results of the expression and association analyses in this study, both the *CAST* MN and *GH* AA types could be considered to be the most important candidate genotypes for better FA composition—lower SFA, omega 6, and omega 3 ratio, higher PUFA content and nutritional value of intramuscular fat. The influence of these genotypes on lamb birth weight and average daily gain was also significant. It is important to note that the GH BB genotype, although it did not have a greater effect on the FA composition of intramuscular fat, especially the content of omega 3 FA, the ratio of omega 6 to omega 3 corresponded to the WHO-recommended ratio of up to 4, and the average daily gain of lambs was the highest of all the studied cases of lamb genotypes. Nevertheless, FA composition is well-defined as a way to describe phenotypic traits, which are possible to improve through genetic selection. The identification of genetic factors controlling FA composition could be implemented in breeding programmes to select animals that produce higher PUFA and lower SFA contents in meat [75,76].

## 4. Conclusions

Based on the research, it can be concluded that the purebred Lithuanian Black-Headed lambs had higher birth weight, better intramuscular fat, and an FA profile associated with lower SFA and higher PUFA contents; therefore, it is not appropriate to cross them with Ile de France, Suffolk or Texel breeds. However, better average daily gain results were obtained by crossing Lithuanian Black-Headed sheep with Ile de France or Suffolk breeds. The genetic influence of the meat-variety breeds was responsible for this, most likely due to positive heterosis, which is advantageous for meat production.

Our research suggests that the most useful to grow lambs who have the BB genotype of the *GH* gene and the MN genotype of the *CAST* gene, because these lambs have the highest birth weight and weight gain. An AA genotype of the *GH* gene and an MN genotype of the *CAST* gene could be applied for genetic selection, improving the quality and nutritional value of ovine intramuscular fat with optimal an omega 6 and omega 3 ratio, lower SFA, and higher PUFA. Our results will allow us to increase the productivity and nutritional value of ovine meat and to meet the demands of consumers in the future.

## Figures and Tables

**Figure 1 animals-15-02992-f001:**
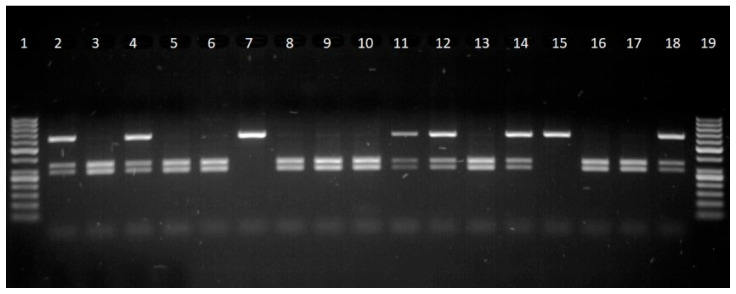
Electropherogram of CAST gene: lanes 1, 19–50 bp DNA ladder, lanes 2, 4, 11, 12, 14, 18—MN genotype, lanes 3, 5, 6, 8, 9, 10, 13, 16, 17—MM genotype, and lanes 7, 15—NN genotype.

**Figure 2 animals-15-02992-f002:**
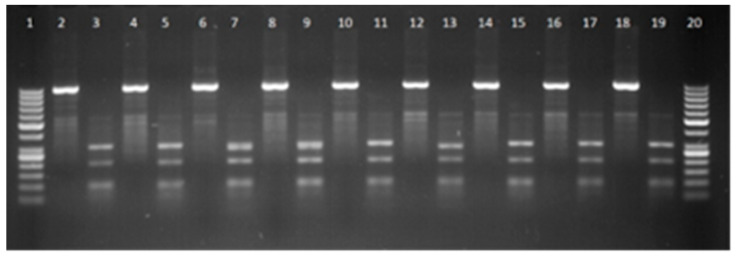
Electropherogram of GH gene: lanes 1, 20–50 bp DNA ladder, lanes 3, 5, 11, 15, 17, 19—AA genotype, lane 13—BB genotype, and lanes 7, 9—AB genotype, 2, 4, 6, 8, 10, 12, 14, 16, 18—PCR product.

**Table 1 animals-15-02992-t001:** Characteristics of *GH* and *CAST* genes primers used for PCR-RFLP analysis.

Gene	Fragment Length, bp	Primer Sequence	Tm	Restriction Endonuclease
*GH*	934	F: GGAGGCAGGAAGGGATGAA	60 °C	HaeIII
		R: CCAAGGGAGGGAGAGACAGA		
*CAST*	622	F: TGGGGCCCAATGACGCCATCGATG	62 °C	MspI
		R: GGTGGAGCACTTCTGATCACC		

**Table 2 animals-15-02992-t002:** Fatty acid composition.

Breed	Lithuanian Black-Headed (a)	Lithuanian Black-Headed * Ile de France (b)	Lithuanian Black-Headed * Suffolk (c)	Lithuanian Black-Headed * Texel (d)
N	43	43	44	45
SFA, % of TFA	43.97 ± 0.63 ***bcd	52.19 ± 0.82 ***a;*d	50.93 ± 0.44 ***a	50.27 ± 0.13 ***a; *b
MUFA, % of TFA	38.36 ± 0.79 *c; ***d	39.03 ± 0.57 ***d	40.32 ± 0.59 *a;***d	43.37 ± 0.17 ***abc
PUFA, % of TFA	15 ± 1.35 ***bcd	8.77 ± 0.87 ***ad	8.75 ± 0.53 ***ad	6.36 ± 0.10 ***abc
OFA (C14 + C16), % of TFA	22.05 ± 0.40 **bd	25.21 ± 1.25 **a	23.36 ± 0.61	23.76 ± 0.34 **a
DFA (UFA + C18), % of TFA	74.50 ± 0.26 ***b;*d	70.56 ± 1.06 ***a;*c;***d	73.14 ± 0.66 *b	73.60 ± 0.33 *a;***b
ω-3, % of TFA	4.52 ± 0.74 ***bd	1.52 ± 0.13 ***acd	2.92 ± 0.43 ***bd	1.02 ± 0.04 ***abc
ω-6, % of TFA	7.25 ± 0.49 ***bcd	4.82 ± 0.34 ***a*c	4.10 ± 0.26 ***a*bd	4.26 ± 0.06 ***ac
ω-6/ω-3	2.76 ± 0.30 ***d	3.29 ± 0.05	2.71 ± 0.31	4.52 ± 0.24 ***a
Trans C18:1 + C18:2, % of TFA	1.26 ± 0.11 ***bc	3.43 ± 0.44 ***acd	0.83 ± 0.07 ***abd	1.69 ± 0.19 ***bc

a, b, c, d—averages, statistically significant difference between in table, marked with different letters (*p* < 0.05). * *p* < 0.05; ** *p* < 0.01; *** *p* < 0.001.

**Table 3 animals-15-02992-t003:** Influence of different breeds on lamb weights.

Breed	N	Birth Weight, kg	Average Daily Gain, kg
Lithuanian Black-Headed a	43	5.70 ± 0.11 ***bcd	0.244 ± 0.01 **b
Lithuanian Black-Headed *Ile de France b	43	5.20 ± 0.07 ***ad	0.266 ± 0.01 **a***d
Lithuanian Black-Headed *Suffolk c	44	4.95 ± 0.17 ***ad	0.268 ± 0.01 *d
Lithuanian Black-Headed *Texel d	45	4.06 ± 0.44 ***abc	0.234 ± 0.01 ***b*c

a, b, c, d—averages, statistically significant difference between in table, marked with different letters (*p* < 0.05). * *p* < 0.05; ** *p* < 0.01; *** *p* < 0.001.

**Table 4 animals-15-02992-t004:** Results of birth weight and weight gain of different genotypes of the CAST gene.

Genotype	N	Birth Weight, kg	Average Daily Gain, kg
MM	100	4.64 ± 0.09	0.250 ± 0.06
MN	75	5.77 ± 0.11 ***	0.253 ± 0.01

*** *p* < 0.001.

**Table 5 animals-15-02992-t005:** Effects of lamb CAST gene genotype on fatty acid profile.

Genotype of *CAST* Gene	MM	MN
N	100	75
SFA, % of TFA	50.46 ± 0.21	48.24 ± 0.72 ***
MUFA, % of TFA	41.29 ± 0.34	39.79 ± 0.50 **
PUFA, % of TFA	8.25 ± 0.37	10.64 ± 0.86 **
OFA (C14 + C16), % of TFA	23.12 ± 0.32	24.33 ± 0.62
DFA (UFA + C18), % of TFA	73.38 ± 0.31	72.55 ± 0.58
ω-3, % of TFA	1.86 ± 0.20	3.03 ± 0.41 **
ω-6, % of TFA	4.66 ± 0.11	5.32 ± 0.33 *
ω-6/ω-3	3.89 ± 0.18	2.76 ± 0.17 ***
Trans C18:1 + C18:2, % of TFA	2.17 ± 0.20	1.06 ± 0.08 ***

* *p* < 0.05; ** *p* < 0.01; *** *p* < 0.001.

**Table 6 animals-15-02992-t006:** Results of birth weight and weight gain of different genotypes of the GH gene.

Genotype	N	Birth Weight, kg	Average Daily Gain, kg
AA a	56	4.75 ± 0.11	0.241 ± 0.01
AB b	61	4.68 ± 0.11	0.233 ± 0.01 ***c
BB c	58	4.78 ± 0.11	0.267 ± 0.01 ***b

a, b, c—averages, statistically significant difference between in table, marked with different letters (*p* < 0.05). *** *p* < 0.001.

**Table 7 animals-15-02992-t007:** Effects of lamb GH gene genotype on fatty acid profile.

Genotype	AA a	AB b	BB c
N	56	61	58
SFA, % of TFA	40.98 ± 0.51 ***bc	50.84 ± 0.49 ***a	49.69 ± 0.47 ***a
MUFA, % of TFA	39.48 ± 0.53 **b	39.99 ± 0.51	41.37 ± 0.51 **a
PUFA, % of TFA	11.54 ± 0.75 *bc	9.17 ± 0.56 *a	8.94 ± 0.86 *a
OFA (C14 + C16), % of TFA	24.41 ± 0.52 **c	24.90 ± 0.73 **c	22.53 ± 0.34 **ab
DFA(UFA + C18), % of TFA	73.12 ± 0.69	71.38 ± 0.69 ***c	74.29 ± 0.27 ***b
ω-3, % of TFA	2.40 ± 0.37	2.42 ± 0.31	2.24 ± 0.42
ω-6, % of TFA	5.42 ± 0.24 **b	4.53 ± 0.18 **a	5.16 ± 0.32
ω-6/ω-3	2.26 ± 0.38 *b**c	3.49 ± 0.31 *a	3.55 ± 0.15 **a
Trans C18:1 + 18:2, % of TFA	1.12 ± 0.22 *b	2.07 ± 0.31 *a	1.54 ± 0.14

a, b, c—averages, statistically significant difference between in table, marked with different letters (*p* < 0.05). * *p* < 0.05; ** *p* < 0.01; *** *p* < 0.001.

## Data Availability

Data is contained within the article.

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
