# Peer review of "Relationship Between Breed, GH and CAST Genotypes, and FA Composition in the Ovine Intramuscular Fat of Musculus Semimembranosus"

_animals, 2025, doi:10.3390/ani15202992_

Round 1

Reviewer 1 Report

Comments and Suggestions for Authors

The manuscript focuses on the polymorphism of two genes associated with sheep production traits, such as lamb growth rate and the fatty acid composition of semimembranosus muscle, in sheep of different breeds. The topic is of interest to the field, but the manuscript cannot be published as is.

The information provided in the introduction is confusing and could be better structured. E.g., by presenting one topic at a time. Please describe the two genes in their genomic context (chromosome, position, gene structure), and their role in the production traits must be clearly defined, at least for the species under consideration. The authors should also explain the rationale for choosing these two genes.

Please consider that the GH gene can be single copy or duplicated in sheep, so there may be two copies of the gene or just one copy, making PCR-based genetic investigations unreliable.

Materials and methods. Please describe briefly the breeds analyzed; it is important to clarify whether they are local or crossbred, and please also give some information about the farm management system and lamb feeding.

Please clarify which SNP was identified through restriction analysis.

It is necessary to indicate the sample collection methods, how the semimembranosus muscle sample was collected, and the relevant statements from your ethics committee.

Please describe the statistical analysis you performed to analyse the results.

The presentation of the results about restriction fragment analysis is unacceptable. Please refer to the actual sequences and indicate which polymorphism was detected.

lines 30-46. The abstract is more of an introduction than a scientific abstract. Please rewrite it, indicating: purpose of the work, animals used and methods, main results obtained, and a brief conclusion.

Some issues need to be addressed; many words are not correctly translated. Please check the English.

Line 64: Please change "regulated" to "encoded".

Line 66: "Several studies have linked the GH gene to". Please explain what characteristic of the GH gene was linked to the listed phenotypes.

Lines 69-70: Please explain, be clearer.

Some Tables are illegible.

Comments on the Quality of English Language

some words are mispelled, e.g. meet instead of meat.

Author Response

Comments 1: [The manuscript focuses on the polymorphism of two genes associated with sheep production traits, such as lamb growth rate and the fatty acid composition of semimembranosus muscle, in sheep of different breeds. The topic is of interest to the field, but the manuscript cannot be published as is. ]

Response 2:[Thank you for your valuable comments and for recognizing the relevance of our study. We carefully revised the manuscript to address the issues you and others reviewer highlighted.]

Comments 2: [The information provided in the introduction is confusing and could be better structured. E.g., by presenting one topic at a time. Please describe the two genes in their genomic context (chromosome, position, gene structure), and their role in the production traits must be clearly defined, at least for the species under consideration. The authors should also explain the rationale for choosing these two genes.]

Response 2:[We have revised this section to improve clarity and coherence by addressing one topic at a time.]

Comments 3: [Please consider that the GH gene can be single copy or duplicated in sheep, so there may be two copies of the gene or just one copy, making PCR-based genetic investigations unreliable.]

Response 3:[We appreciate your insightful comment regarding the GH gene copy number variation in sheep. You are correct that the gene may occur either as a single copy or in duplicated form, which could affect the reliability of PCR-based analyses.If you think it is necessary, we can add a note in the Materials and Methods section, but drawing attention to this potential limitation of PCR-based genotyping.]

Comments 4: [Materials and methods. Please describe briefly the breeds analyzed; it is important to clarify whether they are local or crossbred, and please also give some information about the farm management system and lamb feeding.]

Response 4: [We expanded the Materials and Methods section to provide a clearer description of the animals and management conditions. In particular, we added information on the breeds analyzed.Included a brief description of the farm management system under which the animals were raised. Provided details on lamb feeding practices to give better context for the production environment.]

Comments 5: [Please clarify which SNP was identified through restriction analysis.]

Response 5: [In our study, the polymorphism was identified using restriction fragment length polymorphism (RFLP) analysis with [GH/HaeIII and CAST/MspI ]. In the published literature, this polymorphism is usually described by restriction site and corresponding cleavage pattern, and the exact position of the SNP is not always indicated.]

Comments 6: [It is necessary to indicate the sample collection methods, how the semimembranosus muscle sample was collected, and the relevant statements from your ethics committee.]

Response 6: [We took the comment into account and added this information]

Comments 7: [Please describe the statistical analysis you performed to analyse the results.]

Response 7:[Statistical data analysis was performed using SPSS 25.0 analytical software. The data were presented using descriptive statistics. Normal distributions of variables were assessed using the Kolmogorov–Smirnov test. The one-way analysis of variance (ANOVA) was used to determine statistically significant differences between the means. Multiple comparisons of groups mean were calculated using the Post—Hoc Tukey test. The differences were considered significant at p < 0.05. This information is included in the manuscript]

Comments 8: [The presentation of the results about restriction fragment analysis is unacceptable. Please refer to the actual sequences and indicate which polymorphism was detected.]

Response 8: [In our study, the polymorphism was identified using restriction fragment length polymorphism (RFLP) analysis with [GH/HaeIII and CAST/MspI ]. In the published literature, this polymorphism is usually described by restriction site and corresponding cleavage pattern, and the exact position of the SNP is not always indicated]

Comments 9: [lines 30-46. The abstract is more of an introduction than a scientific abstract. Please rewrite it, indicating: purpose of the work, animals used and methods, main results obtained, and a brief conclusion]

Response 9:[The summary has been rewritten based on comments.]

Comments 10: [Some issues need to be addressed; many words are not correctly translated. Please check the English.]

Response 10:[Thanks for the comments, we have reviewed and corrected the English language]

Comments 11:[Line 64: Please change "regulated" to "encoded".]

Response 11:[Replaced]

Comments 12: [Line 66: "Several studies have linked the GH gene to". Please explain what characteristic of the GH gene was linked to the listed phenotypes.]

Response 12: [Thank you for this helpful observation. We agree that our original wording was too general. In the revised manuscript, we have clarified that the associations reported in previous studies refer to specific polymorphisms within the GH gene. Accordingly, the sentence has been modified to indicate that polymorphic variants of the GH gene have been linked to live weight, body measurements, birth weight, and weaning weight, as reported in the cited studies.]

Comments 13: [Lines 69-70: Please explain, be clearer.]

Response 13: [Information in this section has been supplemented.]

Comments 14: [Some Tables are illegible]

Response 14: [Tables have been reorganized for better understanding]

Comments 15: [Comments on the Quality of English Language: some words are mispelled, e.g. meet instead of meat.]

Response 15: [Thanks for your comments, we have reviewed and corrected the language.]

Reviewer 2 Report

Comments and Suggestions for Authors

Review of  manuscript; Relationship Between Breed, GH and CAST Genotypes and FA Composition in the Ovine Intramuscular Fat of M. Semimembra-Nosus

  1. The abstract must be supplemented with the purpose of the research.
  2. The introduction is written correctly, but the purpose of the research should be added.
  3. Materials and Methods; Please add the age of the mother sheep when the experiment began and ended, and the weight of the lambs at the beginning and end of the experiment. When were the lambings (what time of year)? Were the lambs from twin litters? Were rams or ewes used in the experiment? How were the lambs fed during the experiment, and how were they kept (pasture or indoors)?
  4. The results are described correctly, but the discussion needs to be rewritten, with attention to the practical aspects of the research that are relevant to production.

Author Response

Comments 1:[The abstract must be supplemented with the purpose of the research.]

Response1: [Research objective included ]

Comments 2:[The introduction is written correctly, but the purpose of the research should be added.]

Response2: [Added objective]

Comments 3:[Materials and Methods; Please add the age of the mother sheep when the experiment began and ended, and the weight of the lambs at the beginning and end of the experiment. When were the lambings (what time of year)? Were the lambs from twin litters? Were rams or ewes used in the experiment? How were the lambs fed during the experiment, and how were they kept (pasture or indoors)?]

Response 3:[The methodology has been supplemented by the information we had]

Comments 4:[The results are described correctly, but the discussion needs to be rewritten, with attention to the practical aspects of the research that are relevant to production]

Response 4:[Thank you for your valuable feedback. We have substantially revised the Discussion section to address your concern. In the updated version, we placed greater emphasis on the practical implications of our findings for sheep production, including their potential relevance to growth performance and meat quality traits.]

Reviewer 3 Report

Comments and Suggestions for Authors

The publication is written in clear and accessible English, which, I believe, will be easily understood by specialists and experts in a particular field.

Title

Reflects the topic completely

Abstract:

Very good and concise “Simple summary”.

Overall, the abstract is easy to understand and shows both the beginning of the study and the most important results.

Introduction.

Overall, the Introduction is good, but there are a few places where corrections/clarifications are needed.

Lines 63-64 contain the text: " Growth hormone, a 191-amino acid polypeptide secreted by the somatotropic cells of the anterior pituitary gland, is regulated by the GH (growth hormone) gene". However, it should be noted that the gene does not regulate the protein. The protein is synthesized from the gene sequence.

Starting from line 80, the presence of fat in meat is described. Perhaps it could be supplemented with approximate/average indicators of how much of a particular type of fat is in mutton.

Materials and methods.

Very well-described methods and a good sequence were chosen to show that the study has been carried out sequentially and thoughtfully. But there need same additions:

Were the animals fed the same way and how? The composition of the meat depends on this.

I understand that in previous publications, where the PCR-RFLP method has been used in gene region or, more precisely, polymorphism analysis, fundamental information analysis has not been performed to identify a specific polymorphism, because the database was not yet available. However, the whole sheep genome sequence is currently available, and databases with SNP information are available. Accordingly, it would be necessary to clarify which SNPs are being analysed with PCR-RFLP. It can be done by analysing primer localisation and SNP in region between primers, or by making Sanger sequencing and detecting SNP.

How can you be sure that an analysis was done for GH and/or CAST gene, if there isn’t any information about the position of the analysed region? Did you test the sequences?

I tested the GH primer sequences in NCBI database base on the last sequence (ARS-UI_Ramb_v3.0), I could not find the forward primer in GH gene, but both primers were in region know as Somatotropin or LOC114116958. The region is close to GH hormone, but about 4K bp at a distance. In the Ensemble database there is no information about SNPs, but in the sequence region are 10 HaeIII positions.

Upon further examination of the electrophoresis images, it is evident that there is one SNP in the case of the CAST gene (analysing primer sequences and restriction enzyme possible SNP is rs42261824 G>A); however, in the case of the GH gene, it is difficult to discern from the image in the publication. However, upon examining the cited publication, it is also evident that the PCR fragment is cleaved in several places; the difference lies only in specific fragments, which also suggests the presence of one SNP.

Additionally, examining other publications on GH gene analysis reveals a predominance of genotype A and genotype B, but no specific loci are identified. When searching for information, all such publications mix into a single mass, and it is not clear whether the same SNP is being analyzed or different ones!

Results and discusion

It would be necessary to introduce subsections in the results section to make it easier to keep track of.

The results section is somewhat lacking in presenting the actual results. The text is more of a discussion that does not mention the results obtained in a specific study. The results section should be readable without looking at the table in the middle of each sentence - accordingly, it is necessary also to insert the actual data into the text.

There is no explanation for table 2, what a,b,c,d are. Looking at the Breed names, I assume that the extra letter at the end of the name is a designation. It needs to be clarified.

Also, the abbreviations in the table 2 are not explained.

There aren’t any units of measurement in tables.

I would also recommend turning all tables in vertically, because it is not very easy to read at the moment.

Do the results obtained from the analyzed PCR-RFLP differ between varieties?

In the section where the relationship between genetics and phenotype is analyzed, the text is very unclear. First there is information about the CAST gene and birth weight (Table 4), which should be followed by the CAST gene and fat distribution (Table 5), but a couple of sentences are devoted to it. Immediately, information about the GH region begins (Table 7). And at the end there is GH and birth weight (Table 6). Here we need to clarify.

It may also be necessary to analyze the paragraph division into different fat sections.

In Tables 6 and 7, the genotypes have labels that look like continuations of the genotype. Clarifications are needed.

There should be some kind of summary at the end of the discussion, because currently there is no conclusion in the conclusions section about fat distribution analysis and genotypes.

Considering that the results section does not include an analysis of the distribution of genotypes among the various crosses, it is not immediately clear whether the results obtained in this case do not show the fact why one of the crosses has a better result in terms of weight or fat distribution.

Also, some kind of summary about the significance of the results obtained would be desirable at the end of the discussion. What does the specific result give both to sheep farmers and researchers? How can the obtained results be used further?

Author Response

Comments 1: [Lines 63-64 contain the text: " Growth hormone, a 191-amino acid polypeptide secreted by the somatotropic cells of the anterior pituitary gland, is regulated by the GH (growth hormone) gene". However, it should be noted that the gene does not regulate the protein. The protein is synthesized from the gene sequence.]

Response 1: [Yes, you are right, I corrected the text.]

Comments 2: [Starting from line 80, the presence of fat in meat is described. Perhaps it could be supplemented with approximate/average indicators of how much of a particular type of fat is in mutton.]

Response 2: [Thank you for this valuable suggestion. We have revised the section]

Comments 3: [Were the animals fed the same way and how? The composition of the meat depends on this.]

Response 3: [The study was conducted with animals kept on the same farm, under identical conditions]

Comments 4: [I understand that in previous publications, where the PCR-RFLP method has been used in gene region or, more precisely, polymorphism analysis, fundamental information analysis has not been performed to identify a specific polymorphism, because the database was not yet available. However, the whole sheep genome sequence is currently available, and databases with SNP information are available. Accordingly, it would be necessary to clarify which SNPs are being analysed with PCR-RFLP. It can be done by analysing primer localisation and SNP in region between primers, or by making Sanger sequencing and detecting SNP.]

Response 4: [Thank you for this important observation. In our study, the polymorphisms were identified using PCR-RFLP with the restriction enzymes GH/HaeIII and CAST/MspI. We carefully searched available SNP databases and relevant publications, and we have now clarified in the revised manuscript that the polymorphisms correspond to a C→G substitution in the GH gene and an A→G substitution at position 286 in the CAST gene.]

Comments 5:[How can you be sure that an analysis was done for GH and/or CAST gene, if there isn’t any information about the position of the analysed region? Did you test the sequences?]

Response 5: [We confirm that the analyzes were conducted for the GH and CAST genes using primers and restriction enzymes that have been widely applied and validated in previous studies. The primers used in our study were designed specifically for fragments within the GH and CAST genes and have been published in earlier work. We acknowledge that we did not perform sequencing of the amplified fragments in this study. Therefore, while the targeted regions correspond to those reported in the literature and the observed restriction patterns are consistent with the expected polymorphisms.]

Comments 6: [I tested the GH primer sequences in NCBI database base on the last sequence (ARS-UI_Ramb_v3.0), I could not find the forward primer in GH gene, but both primers were in region know as Somatotropin or LOC114116958. The region is close to GH hormone, but about 4K bp at a distance. In the Ensemble database there is no information about SNPs, but in the sequence region are 10 HaeIII positions.]

Response 6: [Yes, this fragment has 10 HaeIII recognition sequences, our studied polymorphism is the first of the recognition sequences, as mentioned in the literature. ]

Comments 7: [Upon further examination of the electrophoresis images, it is evident that there is one SNP in the case of the CAST gene (analysing primer sequences and restriction enzyme possible SNP is rs42261824 G>A); however, in the case of the GH gene, it is difficult to discern from the image in the publication. However, upon examining the cited publication, it is also evident that the PCR fragment is cleaved in several places; the difference lies only in specific fragments, which also suggests the presence of one SNP.]

Response 7: [Based on our analyses, we cannot precisely specify the SNP rs number in the GH gene due to the limited information in the publication illustration. However, we can confirm that there is one polymorphism (C→G substitution) in the GH gene and one polymorphism (A→G substitution at position 286) in the CAST gene. Therefore, one SNP was detected in both gene regions.]

Comments 8: [Additionally, examining other publications on GH gene analysis reveals a predominance of genotype A and genotype B, but no specific loci are identified. When searching for information, all such publications mix into a single mass, and it is not clear whether the same SNP is being analyzed or different ones!]

Response 8: [I can confirm that the same regions are being analyzed. They have been extensively studied in the literature, but we wanted to link them to meat quality.]

Comments 9:[The results section is somewhat lacking in presenting the actual results. The text is more of a discussion that does not mention the results obtained in a specific study. The results section should be readable without looking at the table in the middle of each sentence - accordingly, it is necessary also to insert the actual data into the text.]

Response 9: [Thank you for your comment. We have taken it into account and updated the results section - it now clearly presents the results obtained in the text, including specific data, so that the section is understandable and without references to tables.]

Comments 10:[There is no explanation for table 2, what a,b,c,d are. Looking at the Breed names, I assume that the extra letter at the end of the name is a designation. It needs to be clarified.]

Response 10: [We have included an explanation below the tables]

Comments 11: [Also, the abbreviations in the table 2 are not explained.]

Response 11: [Thank you for your comment and we have added the unexplained abbreviations in the table to the list of abbreviations.]

Comments 12:[There aren’t any units of measurement in tables.]

Response 12: [We fixed it]

Comments 13:[I would also recommend turning all tables in vertically, because it is not very easy to read at the moment.]

Response 13: [We fixed that too]

Comments 14:[Do the results obtained from the analyzed PCR-RFLP differ between varieties?]

Response 14: [In this study, no comparison between varieties was performed, therefore we cannot provide data on differences between varieties in PCR-RFLP analysis results.]

Comments 15: [In the section where the relationship between genetics and phenotype is analyzed, the text is very unclear. First there is information about the CAST gene and birth weight (Table 4), which should be followed by the CAST gene and fat distribution (Table 5), but a couple of sentences are devoted to it. Immediately, information about the GH region begins (Table 7). And at the end there is GH and birth weight (Table 6). Here we need to clarify.]

Response 15: [We have clarified this part]

Comments 16: [It may also be necessary to analyze the paragraph division into different fat sections.]

Response 16: [We did it]

Comments 17: [In Tables 6 and 7, the genotypes have labels that look like continuations of the genotype. Clarifications are needed.]

Response 17: [We have added explanations under the tables.]

Comments 18: [There should be some kind of summary at the end of the discussion, because currently there is no conclusion in the conclusions section about fat distribution analysis and genotypes.]

Response 18: [We added.]

Comments 19: [Considering that the results section does not include an analysis of the distribution of genotypes among the various crosses, it is not immediately clear whether the results obtained in this case do not show the fact why one of the crosses has a better result in terms of weight or fat distribution.]

Response 19: [Thanks for the note. The aim of this study did not include an analysis of genotype distribution between different crosses, so we cannot directly relate the results obtained to weight or fat distribution indicators between crosses. This may be an interesting direction for further research.]

Comments 20: [Also, some kind of summary about the significance of the results obtained would be desirable at the end of the discussion. What does the specific result give both to sheep farmers and researchers? How can the obtained results be used further?]

Response 20: [We added]

Round 2

Reviewer 1 Report

Comments and Suggestions for Authors

The manuscript has been improved and can be accepted as is, thank you

Reviewer 2 Report

Comments and Suggestions for Authors

I accept the information entered. All comments have been included.

Reviewer 3 Report

Comments and Suggestions for Authors

Thanks to the authors for improving the manuscript of the publication.

Now the article is more complete and easier to read. There is a clear distribution of results and visible analysis.

I would recommend continuing the research in the future, separating the varieties, because this way a better internal distribution of the variety is visible. Especially, considering that the article and the study are studying the Lithuanian local variety.

In addition, I would recommend performing a more thorough bioinformatics analysis of the analyzed polymorphisms in order to provide specific information about the localization and possible functionality of the SNPs in the article.

I agree that the specific two regions have been studied extensively in other populations and so far no one has analyzed the specific SNPs in more depth, but the authors of the article have such an opportunity in future publications.

I recommend the article for publication.